# All-fibre-coupled terahertz single-pixel imaging for biomedical applications

Sen Mou [1], Rayko I. Stantchev[1,2], Sonal Saxena [3], Huiliang Ou[1], Shreeya Rane[1], Sophie L. Pain [4,5], John D. Murphy [4,5], Euan Hendry[3], James Lloyd-Hughes [1] & Emma Pickwell-MacPherson [1]✉

Real-time, non-invasive imaging techniques are essential for advancing biomedical diagnostics and material analysis, yet existing terahertz (THz) systems often suffer from limited speed, bulky designs, and poor adaptability to in situ environments. Addressing these challenges, we present a fully fibre-coupled THz attenuated total internal reflection single-pixel imaging system, offering a compact, flexible, and robust platform for non-destructive spectroscopy and in vivo imaging. This all-fibre architecture enables seamless integration for in situ biomedical applications, including measurements directly on patients. Central to our design is a THz spatial light modulator based on an unpassivated silicon wafer, facilitating high-speed modulation and enabling video-rate imaging with a spatial resolution down to 360 $\mu$m. Despite being in the reflection geometry and using fibre-coupled light, our system achieves an imaging throughput exceeding 30,000 pixels per second for 64-by-64 images - over five-fold higher than the state of the art - representing a substantial improvement in real-time THz imaging capabilities.

Terahertz (THz) light has emerged as a non-destructive and non-ionising alternative to X-rays in spectroscopy and imaging due to its low photon energy (few meV), making it particularly well-suited for biological tissue evaluation. Furthermore, THz exhibits high sensitivity to water content in biological tissues, as water strongly attenuates THz signals. This property enables THz spectroscopy to differentiate between cancerous and normal tissues by exploiting the distinct THz attenuation characteristics resulting from different water content[1–3]. This also provides the capability to monitor transdermal drug delivery processes[4–7], assess skin occlusion effects[8,9], and evaluate the impact of cosmetics on skin hydration[10]. Another advantage of THz spectroscopy is its ability to detect spectral features arising from characteristic rotational and vibrational modes, facilitating its applications in other fields such as agriculture and environmental science. For instance, Martinez et al. demonstrated the identification of fungal infections in chestnuts by leveraging the unique spectral signatures of infected tissues[11]. Mushtaq et al. revealed that THz spectra offer valuable insights into the impact of heavy metal exposure on diatom chain length[12].

THz transmission spectroscopy is widely used for investigating thin films[13,14], dielectrics (e.g., paper, plastics, and ceramics)[15,16], polymers[17], and semiconductors[18]. In contrast, attenuated total reflection (ATR) is particularly advantageous for measuring aqueous samples and other highly absorptive materials. In a typical THz ATR setup, THz light undergoes total internal reflection (TIR) at the prism-sample interface, generating an evanescent field within the sample. Typically, the sample thickness exceeds the penetration depth of the evanescent field, ensuring sufficient attenuation of the evanescent field within the sample. THz ATR has been widely utilised for material characterisation in various applications, including the evaluation of tetracycline hydrochloride solutions[19], DNA molecules[20], and rat brain injuries[21]. Recently, Wu et al. developed a triple internal reflection

[1]Department of Physics, University of Warwick, Coventry CV4 7AL, United Kingdom. [2]Department of Physics, National Sun Yat-Sen University, Kaohsiung, Taiwan. [3]Department of Physics and Astronomy, University of Exeter, Exeter EX4 4QL, United Kingdom. [4]School of Engineering, University of Warwick, Coventry CV4 7AL, United Kingdom. [5]School of Engineering, University of Birmingham, Edgbaston, Birmingham B15 2TT, United Kingdom. ✉e-mail: E.MacPherson@warwick.ac.uk

prism to enhance the sensitivity of THz ATR spectroscopy[22]. Fu et al. introduced a method to apply ATR to samples thinner than the penetration depth. TIR also plays a critical role in manipulating THz polarisation using Fresnel wave retarders[23,24] and phase-compensated mirror-TIR devices[25], controlling THz carrier-envelope phase[26], and improving the modulation depth of THz photomodulators[27,28].

THz imaging has seen growing application in biomedical fields. For instance, Osman et al. developed a handheld portable THz scanner to assess and monitor burn injuries in porcine skin in vivo[29]. Qi et al. employed a THz quantum cascade laser (QCL) confocal imager to differentiate human skin pathologies[30], while Wang et al. applied THz imaging to evaluate the effectiveness of silicone gel sheeting in scar treatment[31]. However, most reported THz imaging systems rely on mechanical raster scanning, which results in slow image acquisition rates. THz computational single-pixel imaging (SPI)[28,32–40] offers a promising solution to achieve high frame rates and spatial resolution without adding significant costs to the overall system. In a THz SPI system, the THz beam is encoded using a series of patterned masks generated by a THz spatial light modulator (SLM). The modulated THz light is then reflected off or transmitted through a sample, after which the signals are detected by a single-element detector, and the image is reconstructed using the known mask patterns and the corresponding THz signals. Mechanical[41,42] and optically controllable[28,43,44] THz SLMs have been proposed. Optically controllable THz SLMs typically use a digital micromirror device (DMD) to encode visible light, which is then projected onto a semiconductor wafer, such as silicon. The modulated semiconductor wafer acts as the THz SLM. The performance of an optically modulated semiconductor SLM is highly dependent on the carrier lifetime[45]. While a longer carrier lifetime enhances the THz modulation depth, it degrades the spatial resolution and the switching speed. Off-the-shelf silicon wafers without passivation typically exhibit short carrier lifetimes (several microseconds), resulting in limited modulation depth. To overcome this limitation, surface passivation techniques[44–47] can be employed to engineer the carrier lifetime, achieving an optimal balance between modulation depth, spatial resolution, and switching rate. For example, Romain et al. developed an electrically tunable THz photomodulator, where the carrier lifetime can be flexibly adjusted via a bias voltage[46].

Despite the promising potential of ATR and THz SPI for biomedical applications, the integration of these two technologies has been hitherto applied only to sub-THz continuous wave sources in free space[34]. Versatile, flexible, all-fibre-coupled broadband THz ATR SPI

has yet to be explored. Here we have developed an all-fibre-coupled design, including the spatial patterning of the broadband THz beam by visible light, making it highly adaptable for the development of compact and portable THz ATR SPI modules that can function as handheld devices[29,48] or be seamlessly integrated with robotic systems[49]. This advanced imaging system will enable real-time, in vivo, and in situ imaging of biological tissues, unlocking transformative potential for applications such as non-invasive skin cancer diagnosis and precision-guided surgical interventions.

## Results
### THz power transmittance
Figure 1a shows the all-fibre-coupled THz ATR SPI system. Figure 1b illustrates the ATR SPI module in detail, and the inset shows the light path inside the prism and silicon wafer. A photograph of the system is shown in Supplementary Fig. 2. The system is built based on a commercial THz time-domain spectrometer. A detailed description of the setup is presented in the section Materials and methods.

The Fresnel coefficients determine the power transmittance. Both polarisation and birefringence affect the Fresnel coefficients. The Fresnel coefficients of s- and p-polarised lights are governed by distinct formulas, and birefringence influences these coefficients through an angle-dependent refractive index. Figure 2a compares the power transmittances of s- and p-polarised THz light. A detailed calculation method is presented in the Supplementary Information. The horizontal axis represents the angle of incidence within the quartz prism. The power transmittance accounts for the dual transmissions at the quartz-silicon interface and a single reflection at the silicon-air interface (see the inset in Fig. 1b). As illustrated in Fig. 2a, the power transmittance of p-polarised light exhibits distinct trends: it decreases to zero as the angle of incidence increases from zero to the Brewster angle at the silicon-air interface, sharply increases from the silicon-air Brewster angle to the silicon-air critical angle, gradually rises to the global maximum at the quartz-silicon Brewster angle, and finally drops to zero as the angle increases from the quartz-silicon Brewster angle to 90°. In contrast, the power transmittance of s-polarised light increases to its maximum as the angle of incidence increases from zero to the silicon-air critical angle, after which it steadily declines to zero between the silicon-air critical angle and 90°.

Both p- and s-polarised light undergo TIR at the silicon-air interface beyond the critical angle. Notably, at the quartz-silicon Brewster angle, p-polarised light undergoes no transmission loss, favouring the

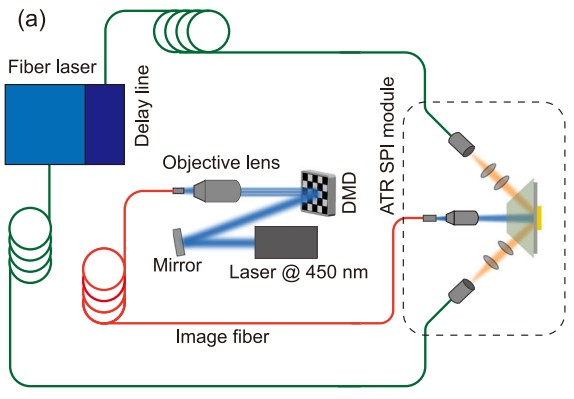
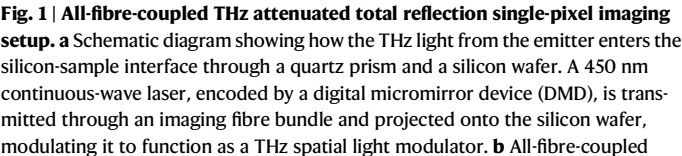

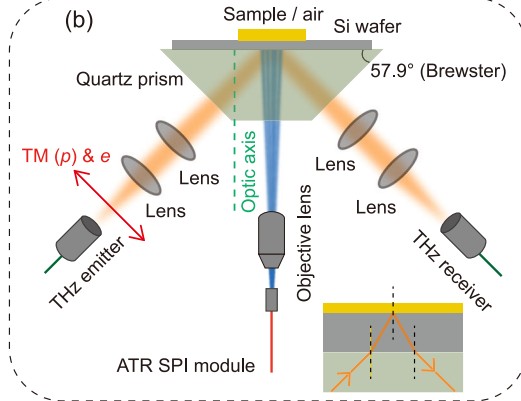

**Fig. 1 | All-fibre-coupled THz attenuated total reflection single-pixel imaging setup. a** Schematic diagram showing how the THz light from the emitter enters the silicon-sample interface through a quartz prism and a silicon wafer. A 450 nm continuous-wave laser, encoded by a digital micromirror device (DMD), is transmitted through an imaging fibre bundle and projected onto the silicon wafer, modulating it to function as a THz spatial light modulator. **b** All-fibre-coupled

attenuated total reflection (ATR) single pixel imaging (SPI) module. The optical axis of the quartz prism is perpendicular to the two parallel surfaces of the quartz prism. The p-polarised THz light is also an extraordinary wave. The prism's base angle of 57.9° equals the Brewster angle for a p-polarised THz light at the quartz-silicon interface from quartz to silicon. The inset shows the light path inside the prism and the silicon wafer.

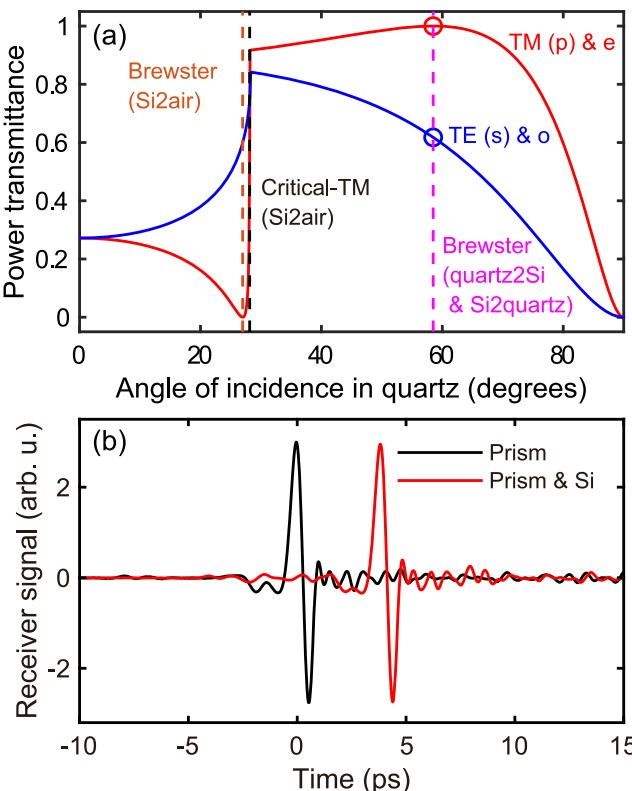

**Fig. 2 | THz power transmittance and receiver signals. a** THz power transmittance. The circle symbols indicate the power transmittance for the experimental prism base angle of 57.9°. THz light propagates through the quartz-silicon interface and is then reflected at the silicon-air interface such that it passes back through the silicon to the quartz (see the inset of Fig. 1b). **b** Experimental receiver signals through the quartz prism with and without a silicon wafer on top of it. The amplitudes of the receiver signals are the same despite the additional dual transmissions at the quartz-silicon interface when a silicon wafer is present. The angle of incidence at the Brewster angle ensures that the additional dual transmissions are lossless.

imaging application. As a result, *p*-polarised THz light is selected in this work, and the prism's base angle is designed to match the quartz-silicon Brewster angle. Figure 2b compares the experimental receiver signals with and without a silicon wafer. The THz receiver signal amplitude remains identical in both cases, confirming the successful dual lossless transmissions of *p*-polarised light through the quartz-silicon interface at Brewster angle and TIR at the silicon-air interface.

### Imaging of cartwheel sample

We imaged a cartwheel sample using a full set of [1 −1] masks. Images are obtained at each temporal point in a 6-ps-wide window with a step of 0.05 ps, a DMD switch rate of 10 kHz, and a 64-by-64 pixel resolution. Each image is an average of 20 image measurements and normalised to a THz beam reference image (more details on the normalisation process are given in the Supplementary Information). We call the resulting intensity the normalised THz signal (see Supplementary Eq. 16) and the unnormalised one the receiver signal, for example, the signal shown in Fig. 2b. In this work, the DMD masks are constructed using a Paley type-I Hadamard matrix, ensuring all masks exhibit similar spatial frequencies. This uniformity enables an easy selection of a subset of masks for undersampling to improve image frame rate[28]. Each 2D mask is flattened into a 1D vector, and the collection of these 1D vectors forms the measurement matrix. Simultaneously, each mask corresponds to a single output from the single-pixel detector, and the detector outputs for all masks collectively form the measurement vector. Image reconstruction is achieved by

multiplying the pseudoinverse of the measurement matrix by the measurement vector[50,51]. The pseudoinverse matrix is calculated based on the regularisation method described in ref. 51.

A gold cartwheel was fabricated on a 2mm-thick z-cut quartz substrate. In the experiment, the cartwheel side attaches to the silicon surface. Edible sunflower oil is employed at the quartz-silicon and silicon-sample interfaces to eliminate air gaps. The THz light incident on the gold cartwheel is reflected back and detected by the receiver. In contrast, because of the Brewster angle, the remaining THz striking the quartz substrate fully transmits into the substrate. THz imaging is performed within a time window centred at the time delay corresponding to the reflection at the silicon-sample interface. Although the THz light transmitted into the substrate experiences total internal reflection and arrives at the receiver, the time delay is outside the time window of interest and thus does not affect the imaging result. As a result, only the THz light incident on the gold cartwheel contributes to the THz imaging.

Figure 3 shows the imaging results of the cartwheel sample. The first row shows the images at the peak (Fig. 3a) and trough (Fig. 3b) in the time domain, and the image at the peak of 0.63 THz (Fig. 3c) in the frequency domain. The inset in Fig. 3a shows the optical image of the cartwheel sample with a red square indicating the THz imaging region. Time-domain images are normalised using a THz beam reference image to eliminate the effects of inhomogeneous THz or blue light distribution. The frequency-domain images are obtained by Fourier transforming the normalised time-domain signal at each pixel. All images show high absolute THz signal values in the metal cartwheel regions, while the quartz regions exhibit very low absolute THz signal values due to the Brewster angle.

A representative pixel on the cartwheel is marked with a circle in the first row. Figure 3d plots the normalised THz signal as a function of time at the marked pixel, with vertical red and blue dashed lines indicating the peak and trough, respectively. These lines correspond to the time delays for the images shown in Fig. 3a, b. Figure 3e plots the spectrum of the THz signal shown in Fig. 3d, with a red dashed line marking the spectral peak. The line corresponds to the frequency for the image shown in Fig. 3c. The solid lines in Fig. 3d, e represent the mean values, and the shaded regions denote the standard deviations across repeated measurements. We also fabricated a knife-edge sample to estimate the resolution, signal-to-noise ratio, and image contrast to characterise the performance of the system. The results (Supplementary Figs. 5, 6) and the method are presented in the Supplementary Information.

### Undersampling and imaging speed

A high frame rate is desirable for real-time THz imaging, and reducing the total number of measurements via undersampling is a widespread strategy to increase the frame rate. The frame rate ($f_{img}$) can be expressed by the following formula:

$$f_{img} = \frac{1}{t_0 + t_{dmd}} = \frac{1}{t_0 + N^2 \times k \times r/f_{dmd}} \tag{1}$$

where $t_0 = 50$ ms is the waiting time to synchronise the DMD, data acquisition card, and probe delay line to detect the THz pulse[28]. $t_{dmd} = N^2 \times k \times r/f_{dmd}$ indicates the DMD active time to obtain an image frame. $N$ represents the pixel resolution, for example, $N = 64$ for a 64-by-64 pixel resolution. $k$ is the number of DMD masks required to construct a measurement mask, where $k = 1$ for the [1 0] mask and $k = 2$ for the [1 −1] mask. $r$ is the undersampling ratio, and $f_{dmd}$ is the DMD switching rate. It is evident from the formula that the maximum frame rate limit is $1/t_0$. When $t_0 \gg t_{dmd}$, $f_{img}$ approaches $1/t_0$. Conversely, when $t_{dmd} \gg t_0$, $f_{img}$ becomes proportional to $1/t_{dmd}$, and thus is proportional to DMD switching rate $f_{dmd}$. Eq. (1) shows that $f_{img}$ decreases with $N$, leading to a difficulty in comparing the performance of an imaging

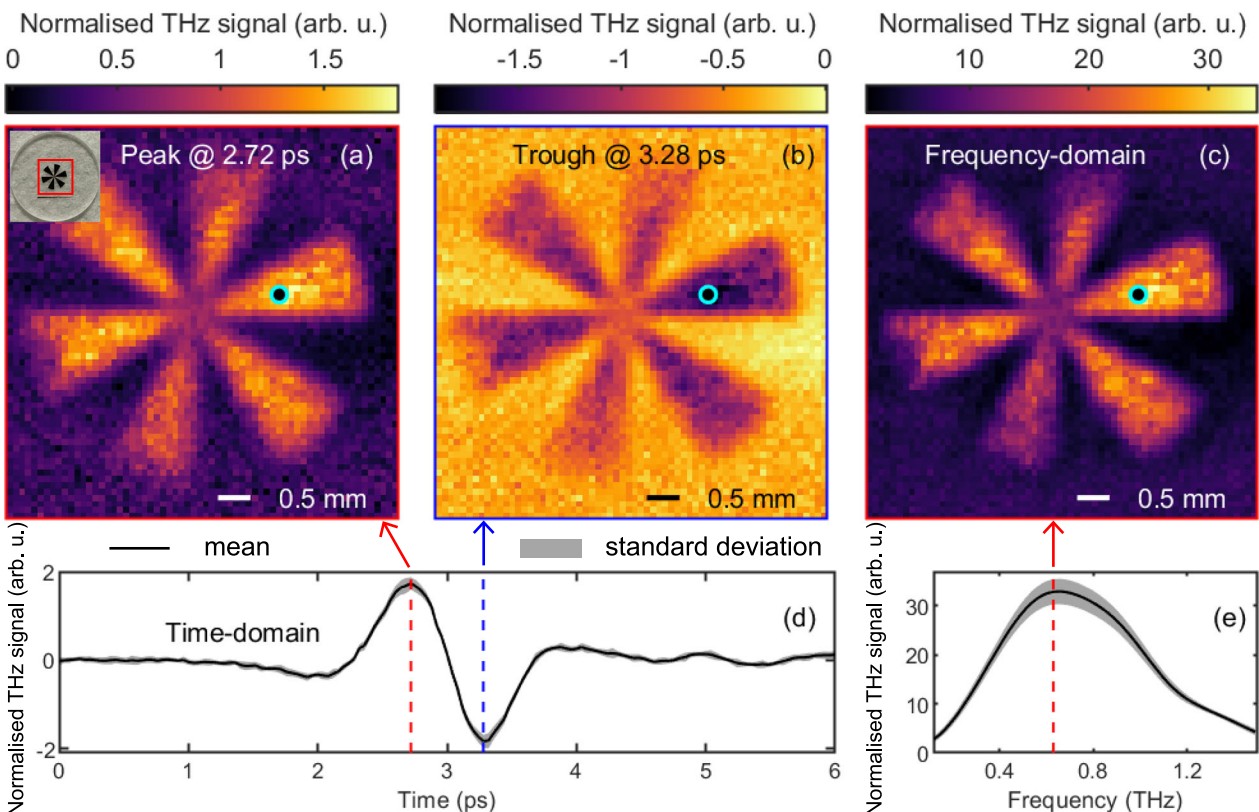

**Fig. 3 | Imaging a gold cartwheel fabricated on quartz with all-fibre-coupled THz attenuated total reflection (ATR) single-pixel imaging (SPI) system.** Time-domain images at the THz peak (**a**) and trough (**b**), and the frequency-domain image at the spectrum peak (0.63 THz) (**c**). The inset in panel (**a**) shows the optical image of the cartwheel, with the red square highlighting the imaging area. The normalised time-domain THz signal and corresponding spectrum at the pixel marked with a black circle in images (**a**–**c**) are shown in panels (**d**, **e**). The solid line represents the mean value, and the shaded region denotes the standard deviation.

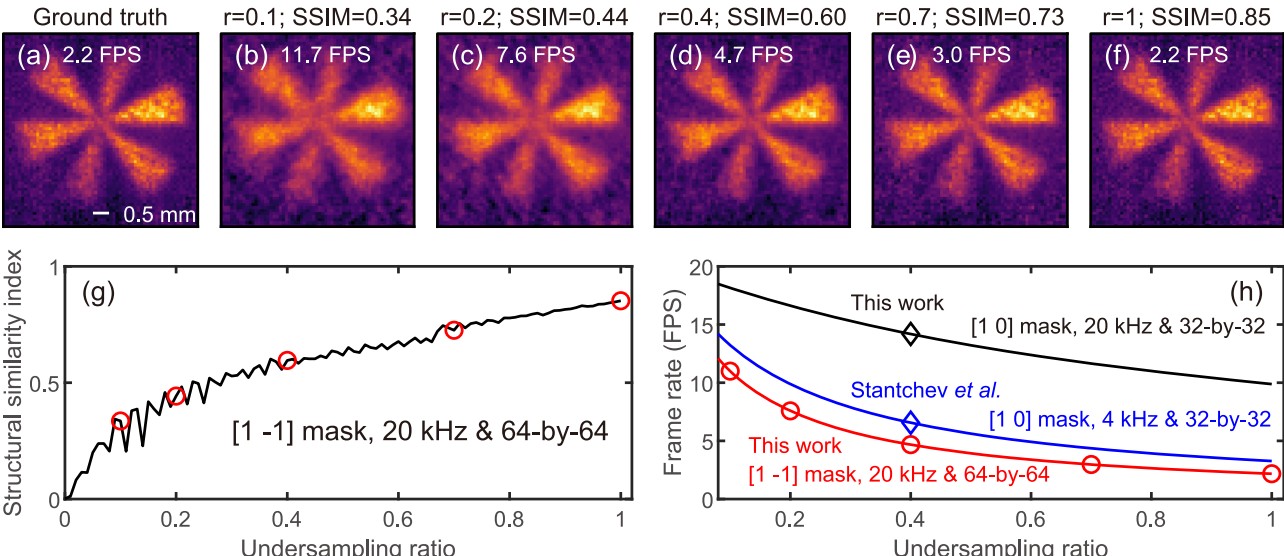

**Fig. 4 | Undersampling and imaging frame rate.** Ground truth image (**a**) and images obtained at different undersampling ratios (**b**–**f**). **g** Structure similarity index (SSIM) versus undersampling ratio. **h** Frame rate versus undersampling ratio. The SSIM and frame rates for the undersampling ratios of images (**b**–**f**) are highlighted with red circles in (**g**, **h**). The standard deviations are not shown in (**g**, **h**) as they are smaller than the symbols.

system for different $N$. In the discussion below, we use instead the number of pixels per second (PPS), $f_{pxl} = N^2 \times f_{dmd}$, to get a fair comparison between different imaging systems[52].

Figure 4 shows the undersampling result using [1 −1] mask and maximum DMD switching rate $f_{dmd}$ of 20 kHz. Figure 4a displays the

ground truth, obtained by averaging 40 measurements using a complete set of [1, −1] masks, corresponding to $r = 1$. Figure 4b–f shows images reconstructed with progressively increasing undersampling ratios from 0.1 to 1, each averaged over 30 image measurements. All the images are normalised to the reference image of the THz beam.

The increasing undersampling ratio reduces the frame rate while improving the structural similarity index (SSIM)[28,53]. Figure 4g illustrates the relationship between SSIM and the undersampling ratio, with the ground truth in Fig. 4a used as the reference image for the calculations. The red circles represent the SSIM values for the images shown in Fig. 4b–f. The SSIM values (0.85, with a full set of masks) are comparable to our previous results[28] (0.81). Measurements were repeated, and standard deviations were found to be smaller than the symbols in Fig. 4g, h. We note that the oscillations shown in Fig. 4g arise from the different sets of measurement masks rather than measurement uncertainty. Although the Paley type-I masks exhibit similar spatial frequencies, they do not show the same spatial frequency. This leads to lower SSIM for a higher sampling ratio, i.e., the oscillations.

Figure 4h shows the frame rate as a function of the undersampling ratio for various combinations of mask type, DMD switching rate, and pixel resolution. The red curve shows the frame rate versus the undersampling ratio for the experimental parameters used for the images in the first row ($N = 64$, $k = 2$, and $f_{dmd} = 20$ kHz). The red circles represent the frame rate of the images shown in Fig. 4b–f. The blue curve shows the frame rate versus the undersampling ratio for the parameters employed by Stantchev et al. ($N = 32$, $k = 1$, and $f_{dmd} = 4$ kHz)[28]. They achieved a frame rate of 6.6 FPS (6,758 PPS) for 32-by-32 imaging at a sampling ratio of 0.4, as indicated by the blue diamond symbol. The short lifetime of the intrinsic silicon wafer used in this work allowed us to utilise the DMD's maximum switching rate of 20 kHz. The black curve shows the frame rate versus the undersampling ratio in this work for $N = 32$, $k = 1$, and $f_{dmd} = 20$ kHz. Compared with the results reported by Stantchev et al.[28], we improved the frame rate to 14.2 FPS (14,540 PPS) at the same sampling ratio of 0.4, as indicated by the black diamond symbol, approaching the maximum frame rate limit $1/t_0 = 20$ FPS. The parameter $t_0$ limits FPS and PPS. This effect on PPS reduces with increasing $N$. We reach 31,049 PPS (7.58 FPS) with $N = 64$, approaching the PPS of the ideal case, that is, $t_0 = 0$. For this case, PPS is 50,000 and independent of $N$. By utilising a single-pixel imaging approach with a rapid DMD switching rate, the PPS exceeds that achievable in mechanically-swept THz QCL imaging systems, typically $f_{pxl} = 2,700$ PPS[30] to 4,100 PPS[54].

## Imaging of porcine tissue

We imaged a porcine tissue to demonstrate the sensitivity of the all-fibre-coupled THz ATR SPI system to biomedical tissues. A full set of [1 −1] masks, a DMD switch rate of 10 kHz, and a 64-by-64 pixel resolution were employed. We covered the tissue with cling film to prevent dehydration during the experiment. Images are obtained at each temporal point in a 6-ps-wide window with a step of 0.05 ps. Each image is an average of 20 measurements. Reference images of the THz beam were obtained with a bare silicon wafer using the same experimental parameters to normalise time-domain images, e.g., Fig. 5a.

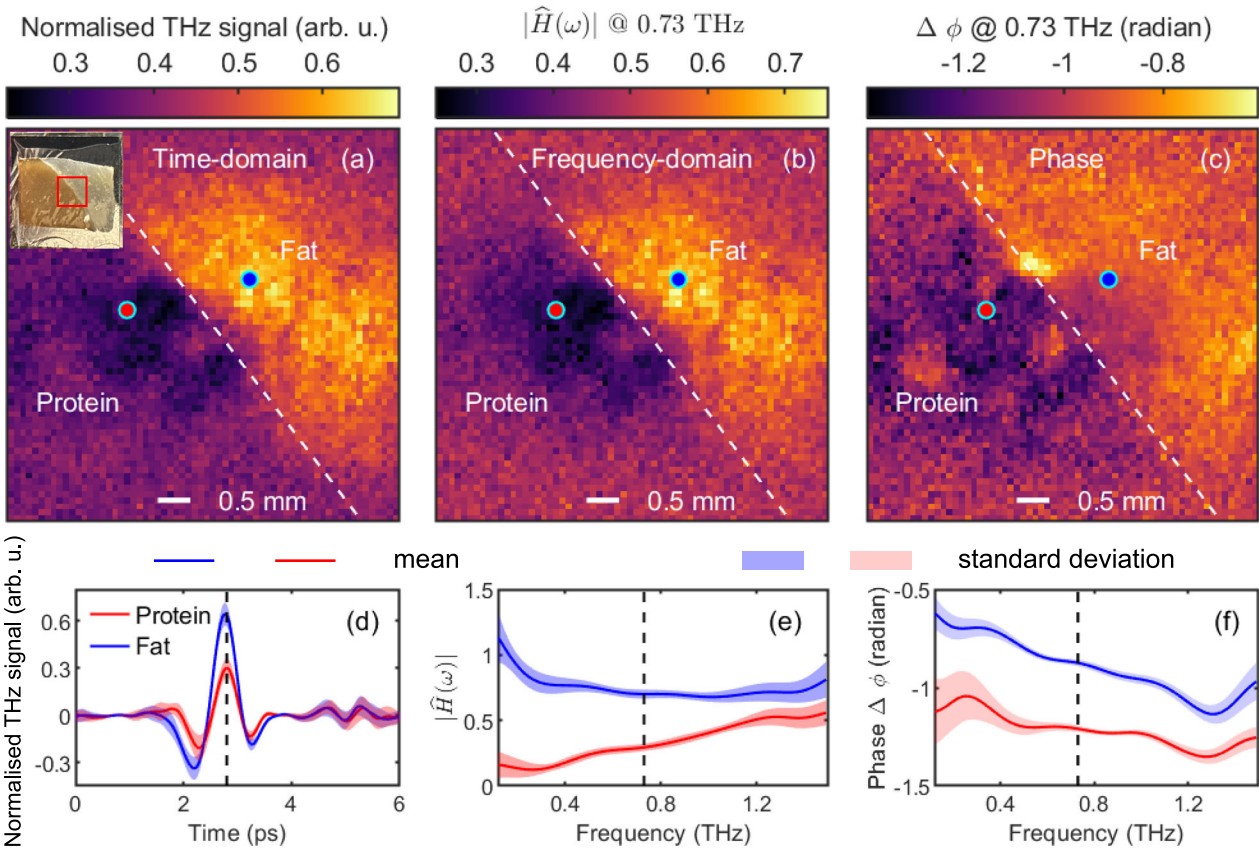

**Fig. 5 | Application of all-fibre-coupled THz attenuated total reflection (ATR) single-pixel imaging (SPI) system for distinguishing fat and protein in porcine tissue.** The pixel resolution of the images is 64 × 64. **a** Time-domain image at the THz signal peak. The inset shows the porcine tissue with a red square indicating the THz imaging area. **b** The magnitude of the complex transfer function ($|\hat{H}(\omega)|$). **c** The argument of the complex transfer function ($\arg(\hat{H}(\omega)) = \Delta\phi(\omega)$). The white dashed lines in the first row serve as a visual guide showing the boundary between the protein and fat. The time-dependent normalised THz signals (**d**), frequency-dependent $|\hat{H}(\omega)|$ (**e**), and phase difference (**f**) at the fat and protein representative pixels marked with blue and red circles in images shown in the first row. Dash lines in (**d**–**f**) indicate the time (**a**) and frequency (**c**, **d**) positions for the images in the first row. The solid lines in (**d**–**f**) represent the mean value, and the shaded regions in (**d**–**f**) denote the standard deviations.

Receiver signals of the reference and sample at each pixel are Fourier transformed. We define the ratio between the Fourier transforms as a complex transfer function $\hat{H}(\omega)$, which can be written as:

$$\hat{H}(\omega) = \frac{S_{smp}(\omega)}{S_{air}(\omega)} = \frac{r_{smp}(\omega)}{r_{air}(\omega)} \qquad (2)$$

where $S_{smp}(\omega)$ and $S_{air}(\omega)$ are the Fourier transforms of the sample and reference time-dependent receiver signals, $\omega$ is the THz angular frequency, $r_{smp}(\omega)$ and $r_{air}(\omega)$ are Fresnel reflection coefficients at the silicon-sample and silicon-air interfaces. The time-domain receiver signal is the convolution of the THz electric field and the response function of the receiver. In the frequency domain, the receiver signal is given by the product of the THz electric field and the response function, i.e., $S_i(\omega) = E_i(\omega)F_i(\omega)$, and $F_i(\omega)$ is the receiver function. Although $F_i(\omega)$ cancels in Eq. (2), it is not expected to cancel completely for our normalised images, such as Fig. 3c, as we sum over the frequency spectrum during normalisation to improve signal-to-noise ratio. However, we note that this frequency summation does not qualitatively affect the resulting images, which suggests the frequency dependence of $F_i(\omega)$ is weak in the frequency range of this work. Thus, one can roughly assume the receiver signal is proportional to the THz electric field.

The magnitude $|\hat{H}(\omega)|$ illustrates the amplitude ratio of the THz signal. The argument $\arg(\hat{H}(\omega))$ stands for the phase difference between $S_{smp}(\omega)$ and $S_{air}(\omega)$, i.e., $\Delta\phi(\omega) = \arg(\hat{H}(\omega)) = \arg(S_{smp}(\omega)) - \arg(S_{air}(\omega))$. Fig. 5a shows the normalised time-domain image of a porcine tissue sample at the peak. Figure 5b, c plot the images of $|\hat{H}(\omega)|$ and $\arg(\hat{H}(\omega))$, i.e., $\Delta\phi(\omega)$ at 0.73 THz. The inset in Fig. 5a presents the optical image of the sample with a red square indicating the THz imaging region. This region comprises protein and fat separated by a diagonal boundary. In the first-row images, the dashed white diagonal line marks the boundary between protein and fat. The red and blue circles mark the representative points in the protein and fat regions. The second row illustrates the normalised time-domain THz signal (Fig. 5d), $|\hat{H}(\omega)|$ (Fig. 5e) and $\Delta\phi(\omega)$ (Fig. 5f) for the representative points marked in the protein and fat regions. The solid lines in Fig. 5d–f represent the mean values, and the shaded regions denote the standard deviations. The black dashed lines in the second row highlight the time and frequency points for the images in the first row. The spectra shown in Fig. 5e, f clearly distinguish between protein and fat across the full frequency range. Similarly, the time-dependent normalised THz signal curve in Fig. 5d differentiates protein and fat, except at the zero-crossing points.

### In vivo real-time imaging of scab on wound
We used the all-fibre-coupled THz ATR SPI system to image a scab and normal skin on a volunteer's arm, demonstrating its capability to assess skin in vivo. The imaging is carried out at the peak of the THz pulse trough with a full set of [1 −1] masks, a DMD switch rate of 20 kHz, and a 64-by-64 pixel resolution, leading to 2.2 FPS (8912 PPS). The images were averaged over 20 measurements and normalised with the same method shown in the Supplementary Information. Figure 6a shows the resulting THz image of the scab together with a photo (inset). The THz image successfully distinguished the scab and the surrounding healthy skin. It also correctly depicts the shape of the scab shown in the optical photo. The scab region exhibits a stronger reflected THz signal (darker colour in Fig. 6a) compared with the surrounding healthy skin because the scab is drier and therefore gives a greater reflection coefficient than the healthy skin. Figure 6b shows the THz image of normal skin, which attenuates the THz signal homogeneously to a level comparable to the healthy surrounding skin shown in Fig. 6a. Figure 6c shows the normalised THz beam for comparison. We also recorded a video (see Supplementary Movie 1) to illustrate the measurement of the scab and demonstrate the ability of the proposed system for real-time in vivo applications.

### Discussion
We have developed an all-fibre-coupled THz ATR SPI system that successfully integrates the synergistic benefits of all-fibre coupling, ATR, and SPI. Firstly, the all-fibre-coupled design enables the development of a flexible and compact THz ATR SPI module that can act as a handheld device or be integrated with a robot. This facilitates in situ and in vivo imaging of biological tissues, which is unachievable with transmission-based geometries. We have demonstrated this potential by using the prototype to successfully differentiate between the protein and fat in a porcine tissue sample. We also successfully demonstrated the capability of the system for real-time in vivo biomedical applications by applying the system to image a scab on a wound. Additionally, SPI allows a high imaging rate, making it suitable for real-time applications. The THz SLM's short carrier lifetime of 7 μs enables the highest DMD switching rate of 20 kHz to be utilised, achieving 31,049 PPS with 64-by-64 resolution using [1 0] mask and an undersampling ratio of 0.4. The short lifetime also leads to a high resolution of 360 μm ($\lambda/1.4$ at the THz spectrum peak of 0.6 THz). As a result, the all-fibre-coupled THz ATR SPI system significantly advances the practical implementation of THz technology for in situ, in vivo and real-time imaging of biological tissues for biomedical applications such as the diagnosis and surgical removal of skin cancers.

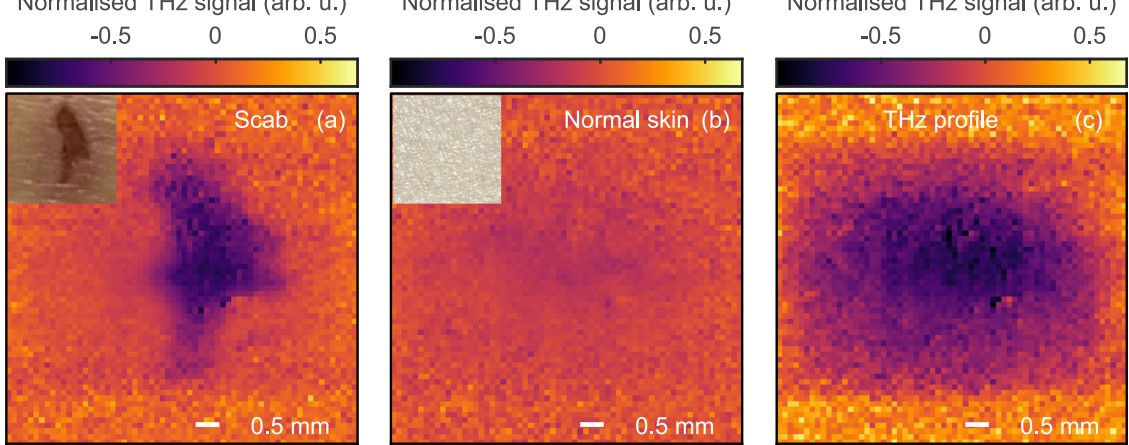

**Fig. 6 | In vivo real-time imaging of scab on wound at THz trough in time domain. a** THz image of the scab. **b** THz image of normal skin for comparison. **c** Normalised THz profile. Insets in panels (**a**, **b**) are the optical photos of the scab and normal skin in measurement.

## Methods

### Experimental setup

The setup is based on a commercial THz time-domain spectrometer (Menlo Terasmart). As shown in Fig. 1, a fibre-based femtosecond laser delivering femtosecond pulses with a central wavelength of 1550 nm and a repetition frequency of 100 MHz was applied to generate and detect THz light. The THz spectrometer integrates a delay line to vary the time delay between the THz pulse and the probe laser pulse. The power of the laser delivered to the fibre-coupled photoconductive emitter (Fraunhofer TERA 15-TX-FC) and receiver (Fraunhofer Tera15-RX-FC) is around 20 mW. The emitter is biased with 100 V DC, emitting THz with a power of 84 $\mu$W. The signal from the receiver is amplified with a transimpedance amplifier (Femto LCA-S10) and then read by a data acquisition (DAQ) device (NI USB-6351). THz light from the emitter is collimated by a TPX lens and focused by another TPX lens through a quartz prism and a silicon wafer to the silicon-sample interface. THz light reflected from the silicon-sample interface is collimated by a TPX lens and focused by another TPX lens to the receiver. The detected THz pulse exhibits a pulse duration of 0.94 ps, with a spectrum peaked at 0.63 THz and a full-width-half-maximum of 0.78 THz.

A diode laser (Lasertack LAB-450-5000) generates continuous-wave 450 nm-wavelength blue light. The blue light is encoded with a DMD (Vialux V4395) to imprint programmed patterns onto it. A multi-core image fibre (Fujikura FIGH-100-1500N) transmits the patterned blue light from the input to the output. Subsequently, the blue light is projected onto the silicon surface through the quartz prism by an objective lens. The projected laser patterns modulate the silicon wafer, generating carrier patterns replicating the same profile as the incident light. The image fibre comprises 100,000 picture elements and exhibits a large core diameter of 1.4 mm. We applied a 10× microscope objective lens to match the image size from the active area of the DMD to the area of the fibre bundle and also to match the numerical aperture of the lens to the acceptance angle of the fibres. A more complete description of the optimisation of the fibre projection system is presented in the Supplementary Information. The total optical transmission of the fibre projection system for the blue light is 21%, allowing the delivery of 882 mW blue light to the silicon wafer from a laser output of 4.2 W. The beam size on the silicon wafer is 8 mm × 8 mm, resulting in a pump fluence of 13 mW/mm$^2$.

Carrier lifetime plays a vital role in modulation depth, switching speed and image spatial resolution[45]. Hereby, we used an unpassivated 200-$\mu$m-thick, double-side polished, (100) float-zone silicon wafer with a carrier lifetime of 7 $\mu$s (the calculation of the carrier lifetime is explained in the Modulation Depth section of the Supplementary Information) and resistivity greater than 10,000 $\Omega \cdot$ cm. This high-resistivity silicon wafer with a low effective lifetime allows us to achieve high resolution (360 $\mu$m) single-pixel imaging with a high frame rate supported by the highest DMD switching rate (20 kHz) and a THz modulation depth of 12%. While this may seem like a low modulation depth, it works well in practice because the switch-off time is shorter than if a higher modulation depth were used. To increase the modulation depth we would need to use a wafer with a longer carrier lifetime, which would improve signal to noise, but at the cost of imaging time and poorer resolution. Therefore, using the short effective lifetime of 7 $\mu$s allows us to utilise the highest DMD switching rate to achieve high imaging speed. It also gives rise to high imaging resolution, representing a good trade-off between modulation depth, imaging speed, and resolution.

### [1 0] & [1 −1] masks

SPI can be carried out with the mask [1 0] or [1 -1]. The mask [1 0] uses the measured difference in reflection between photo excitation and no photo excitation to reconstruct the image. In comparison, the mask [1 −1] uses the measured difference in reflection between a photo excitation mask [1 0] and its inverse [0 1]. Experimentally, the mask [1 −1] is performed by projecting a [0 1] mask immediately following a [1 0] mask. The signals were then subtracted to give a signal equivalent to using a [1 −1] mask. The subtraction procedure is adopted because a [1 −1] mask cannot be directly projected as the DMD micromirrors have only two states, i.e., on (1) and off (0).

### Reporting summary

Further information on research design is available in the Nature Portfolio Reporting Summary linked to this article.

## Data availability

Data supporting the findings of this study are openly available in Figshare at https://doi.org/10.6084/m9.figshare.30719816.

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

## Acknowledgements

The authors acknowledge support from the Engineering and Physical Sciences Research Council (EPSRC) (EP/V047914/1). SLP is supported by the Royal Academy of Engineering Research Fellowship scheme (RF-2324-23-197). RIS acknowledges the support of the Yushan Young Fellow award (MOE-112-YSFMS-0009-001-P1).

## Author contributions

S.M. performed the experiments with help from R.I.S. and S.R. S.M. and R.I.S. analyzed the data with help from S.R. S.M., S.S., and R.I.S. constructed the experimental setup with help from E.H., H.O., and S.R. S.M. wrote the manuscript with input from S.S., E.H., J.L.H., and E.P.M. All authors checked the manuscript. H.O. fabricated the cartwheel sample. S.L.P. and J.D.M. provided the silicon photomodulators. E.P.M., J.L.H., and E.H. supervised the project.

## Competing interests

The authors declare no competing interests.
