## [Transparent Peer Review file · Nature Communications]

All-fibre-coupled terahertz single-pixel imaging for biomedical applications

Corresponding Author: Professor Emma Pickwell-MacPherson

Version 0:

Reviewer comments:

Reviewer #1

(Remarks to the Author)

his work reports a single-pixel imaging system that combines fiber-coupled visible light modulation with a terahertz (THz) attenuated total reflection (ATR) geometry. By using a digital micromirror device (DMD) to modulate light patterns transmitted to a silicon wafer surface via imaging fiber bundles, the system achieves spatial modulation of terahertz waves. The authors demonstrated its imaging capabilities on porcine tissue samples and human skin wound areas, claiming advantages such as 'all-fiber-coupled', 'real-time imaging', and 'high resolution'.

The results are solid, experiments are properly executed, and it delivers good spatial resolution and frame rates.

Supplementary materials also include theoretical analyses and modeling derivations. However, considering terminology accuracy, technological novelty, and biomedical applicability, I do not think this manuscript meets Nature Communications' high standards for originality and broad impact in the field. My detailed concerns are as follows.

1. Overclaims.

(i) The term 'all-fiber-coupled' in the title and text is inaccurate and misleading. This is because,

(1) Only 450 nm visible light patterns are transmitted via imaging fibers.

(2) THz wave generation, propagation, modulation, and detection rely entirely on free-space optics (TPX lenses, ATR prism, free-space-coupled THz-TDS system).

(3) The 'fiber-based' component is limited to visible light modulation; the primary THz signal path remains free-space. No fiber-compatible THz source/detector is proposed.

This contradicts the common understanding of 'all-fiber-coupled' in THz/optical imaging and may mislead readers into assuming system portability or encapsulation feasibility.

(ii) Additionally, the terms 'in vivo', 'in situ', and 'real-time imaging' are overstated. The system can only detect skin or air-exposed tissues due to its fundamental limitation: modulation requires direct contact between the silicon wafer and the target. This restricts its applicability for genuine biomedical scenarios where internal body operation is essential. In terms of 'real-time imaging', the 2–7 fps frame rate is relatively slow.

2. Engineering Integration Over Core Innovation. Key components are taken from existing technologies:

(1) DMD + silicon wafer for optical THz modulation is a common design.

(2) Hadamard encoding and [1 -1] masking are conventional single-pixel imaging methods.

(3) ATR structures are not new to THz applications.

(4) Image reconstruction uses basic normalized linear superposition, lacking advanced compressive sensing or deep learning approaches.

The performance improvements (2–7 fps frame rate, ~100 μm resolution) stem from component optimization, not fundamental methodological or physical breakthroughs.

3. Limited Imaging Validation and Clinical Relevance

Despite the authors demonstrated the method by imaging porcine tissues and human skin wounds:

(1) Sample diversity is insufficient to represent broad biomedical scenarios.

(2) Results are qualitative; no quantitative comparison exists against clinical standards (e.g., OCT, infrared thermography, ultrasound).

(3) No histopathological registration confirms imaged structures' physiological significance.

(4) Statistical evaluations (SNR, CNR, recognition rates) or cross-sample comparisons are absent.

4. Low Modulation Efficiency Limits Performance

Supplementary material reveals a modulation efficiency of ~12%, with only 21% of initial blue light energy reaching the silicon wafer due to losses in fibers/lenses. This low signal contrast necessitates strong normalization and background suppression, potentially compromising imaging depth and stability under low-SNR conditions.

In conclusion, this work demonstrated solid engineering capabilities in system implementation and offers a reference for THz imaging integration. However, its contributions remain confined to optimizing existing components rather than delivering field-leading innovation. Coupled with misleading terminology, overclaims, narrow application validation, and overstated clinical significance, the manuscript falls short of Nature Communications' standards.

Reviewer #2

(Remarks to the Author)

Dear Authors,

I was asked to review your manuscript NCOMMS-25-44759 "All-fibre-coupled terahertz single-pixel imaging for biomedical applications".

In the manuscript you describe the results obtained with a fibre-coupled terahertz single-pixel imaging system.

A terahertz source is spatially modulated using an optically controlled silicon SLM.

The telecom laser for the terahertz source and detector is fibre coupled and the THz frequency is imaged using TPX lenses, the blue light, used for modulation is fibre coupled using an imaging fibre.

Your work improves on previous experiments in two ways:

1. The shorter lifetimes of carriers in your silicon SLM allows for faster modulation and ultimately higher frame rates.
2. The demonstration of an all fibre coupled experimental setup paves the way for integration of the method onto handheld devices.

I am convinced that your evaluation of your data is valid and your collected data is robust and I can't find any technical flaws with your manuscript.

However, I want to recommend to comment on measurement uncertainties in your manuscript.

As far as I understand, most of your measurements are averaged results, it should be straight forward to include confidence intervals/standard deviations in your plots in figures 3-5.

Your work improves on previous results by an increased frame rate and integrating your setup in an all fibre-coupled manner, which means the inclusion of an imaging fibre.

The telecom fibre for the THz source and detector were already fibre coupled in previous work.

The core methodology of your experiment, however, stays identical.

I still want to argue that advancing technology in this way is still significant, especially in applied sciences, where the step from the lab into the field often can be challenging.

Your presentation is very clear and easily understandable.

I only have a few suggestion for improvements on your manuscript.

1. As already mentioned a more detailed analysis of errors.
2. You are not comparing the structural similarity (SSIM) with your previous work. I would argue that SSIM and FPS/PPS play an important role when benchmarking your experiments against each other.
3. Fig 5c seems to be taken at lower resolution. Why is that?
4. Formatting of formulas in the supplement is often unconventional (\times is often used).

Otherwise, I have no concerns for your manuscript being published in Nature Communications.

Reviewer #3

(Remarks to the Author)

The authors present an interesting article on the intriguing combination of multiple techniques for a useful THz imaging system. The authors also present rudimentary measurements on biological samples, which round the paper off nicely. The methodology seems sound and the information provided is sufficient.

However, the structure of the text does not correspond to that of the journal. The 'Results' section contains a great deal of detail about the setup and methods, which should be found in the 'Methods' section.

The discussion of the effect of THz polarisation on resolution anisotropy is not fully convincing and is not supported by references, simulations or other evidence. Without this, it is merely an assumption and, as such, should not be discussed in such detail.

The core achievement is the ability to image at high resolution and high measurement rates. While the resolution target presented provides a rough idea of the imaging capabilities, some aspects are unclear.

- Resolution: The measurements with the resolution target (Siemens star) indicate varying resolution throughout the image. Given the provided data, this could be due to the orientation of an edge (similar to astigmatism) or the position within the field of view.

- Contrast: The achievable contrast in this image is not provided. It is stated that the background is zero, which would result in infinite contrast; this is certainly not the case.

- Homogeneity: The images of the resolution target show a highly non-homogeneous image in terms of contrast, signal strength and resolution. While this is not surprising, it needs to be quantified. Is it possible to partially compensate for this inhomogeneity by calibrating each pixel?

- Noise: The image noise has not been quantified. This can occur between neighbouring pixels that are expected to have the same signal strength, as well as between subsequently recorded images.

I suggest applying the razor blade method as a simple experiment to quantify the above values. Place a razor blade, or another sharp, well-defined edge, at the sample plane, then take images at different blade positions. Rotate the blade by 90 degrees and repeat. The images can then be compared with the well-known and simple ground truth and the values above can be extracted. I am willing to help if the authors would like to discuss this in more depth.

Other comments:

Line 103. What does 'short' refer to? Please provide an order of magnitude.

Line 128 and following: This focuses on the setup and details rather than the results.

Line 163 and following: This paragraph seems to explain fundamental optics; I suggest shortening it drastically.

Line 179: Why 'also'?

Line 206 and following: Why are both polarizations discussed in detail when the experiment uses just one of them?

Line 297: The e-field is certainly not zero.

Line 299: See above.

Fig. 3c). The plots have the unit 'electric field', but this is imprecise because it is not clear where this electric field is located. I believe the authors plotted the receiver signal, which is not fully linear in the electric field due to the receivers' frequency-dependent roll-off. Furthermore, the setup will have some frequency dependence due to the optics involved and their adjustment. Therefore, for d) and e), I suggest using the term 'receiver signal', and for a to c, I recommend normalising to a full metal mirror as the sample and using the term 'image contrast' or similar. The same applies to other pictures later in the manuscript.

Line 490 and following: Please revise the naming of the components used. If you are unsure, please consult the manufacturer for the correct terminology.

Version 1:

Reviewer comments:

Reviewer #1

(Remarks to the Author)

The authors addressed most of my concerns. However, it is recommended to provide some quantitative data to support the claim that '12% is a nearly optimal trade-off between signal-to-noise ratio and imaging resolution/speed'.

Reviewer #2

(Remarks to the Author)

Dear Authors,

thank you for addressing my concerns with your first manuscript.

I feel that any issues I raised have been resolved and have no concern for your manuscript to be published in Nature Communications.

Reviewer #3

(Remarks to the Author)

Thank you to the authors for the revised manuscript and for addressing all of my comments so comprehensively. Thank you also for the additional knife-edge measurements. I believe these are highly valuable for evaluating such an imaging system. The methodology is sound, and based on the provided data, the obtained values for resolution, SNR and contrast seem plausible. While the interpretation of the anisotropy of the imaging resolution (astigmatism) does not provide a definitive answer, it does offer possible explanations. It would have been great to see a calculation or simulation of this effect, since resolution is one of the key factors of any imaging system. However, this could be the subject of a follow-up paper or letter in a more technical journal.

I have no objections to the publication of this manuscript.

Response to Reviewers

We thank the reviewers for their feedback on our paper. We address all the points below on a point-by-point basis. Our responses are in **blue font**. We have done the experiments suggested and agree that adding these into the paper make it stronger.

Reviewer #1 (Remarks to the Author):

This work reports a single-pixel imaging system that combines fiber-coupled visible light modulation with a terahertz (THz) attenuated total reflection (ATR) geometry. By using a digital micromirror device (DMD) to modulate light patterns transmitted to a silicon wafer surface via imaging fiber bundles, the system achieves spatial modulation of terahertz waves. The authors demonstrated its imaging capabilities on porcine tissue samples and human skin wound areas, claiming advantages such as 'all-fiber-coupled', 'real-time imaging', and 'high resolution'.

The results are solid, experiments are properly executed, and it delivers good spatial resolution and frame rates. Supplementary materials also include theoretical analyses and modelling derivations. However, considering terminology accuracy, technological novelty, and biomedical applicability, I do not think this manuscript meets Nature Communications' high standards for originality and broad impact in the field. My detailed concerns are as follows.

1.1. Overclaims.

(i) The term 'all-fibre-coupled' in the title and text is inaccurate and misleading. This is because,

(1) Only 450 nm visible light patterns are transmitted via imaging fibers.

(2) THz wave generation, propagation, modulation, and detection rely entirely on free-space optics (TPX lenses, ATR prism, free-space-coupled THz-TDS system).

(3) The 'fiber-based' component is limited to visible light modulation; the primary THz signal path remains free-space. No fiber-compatible THz source/detector is proposed.

This contradicts the common understanding of 'all-fibre-coupled' in THz/optical imaging and may mislead readers into assuming system portability or encapsulation feasibility.

Answer: We thank the reviewer for their comments but would like to point out where they have misunderstood our THz set up and why we disagree with their opinion that we are overclaiming. We point out that there is fibre coupling in ALL the THz emission/detection pathways as well as in the optical illumination pathway. By "all-fibre-coupled" we mean that the THz emitter and detector in the imaging head (Fig. 1b) are fibre-coupled to the fiber laser as well as the light modulation path, so points (1-3) of comment 1.1 are incorrect. This makes the imaging head portable and ultimately it will be compatible with a robot arm. The reviewer is right in that the optics inside the imaging head are free-space optics, but this is the part where the THz light meets the skin so it has to be. The point is that the laser light from the fiber laser (to the THz emitter and detector) and from the UV laser (after passing through the DMD and objective lenses) is fiber-coupled. These fibre-coupled THz antennas are well accepted as fibre-coupled optics in the THz community to distinguish them from the free-space sources such as ZnTe, LiNbO₃ and two-colour plasma. Indeed, the fibre-coupled antennas are widely employed to design portable THz

imaging systems, such as the device presented in Ref. 29 (Adv. Photonics Res. 3, 2100095 (2022)).

Reviewer 2 supported the above justification and appreciated the “all-fibre-coupled” idea, commenting on this aspect positively: “**Your work improves on previous results by an increased frame rate and integrating your setup in an all fibre-coupled manner, which means the inclusion of an imaging fibre.**”

(ii) Additionally, the terms 'in vivo', 'in situ', and 'real-time imaging' are overstated. The system can only detect skin or air-exposed tissues due to its fundamental limitation: modulation requires direct contact between the silicon wafer and the target. This restricts its applicability for genuine biomedical scenarios where internal body operation is essential. In terms of 'real-time imaging', the 2–7 fps frame rate is relatively slow.

Answer: The term “in vivo” simply means that the subject being measured is living without having to be excised or sliced to be thin enough for a transmission measurement, which is the exact reason for the reflection configuration. As demonstrated in Fig. 6, we succeeded in applying our imaging system to measure living human skin. The term “in situ” is closely related to the “all-fibre-coupled” idea. The imaging head is all-fibre-coupled so that it is portable and can be directed to the region of interest. The system intrinsically exhibits the “in situ” attribute as it is “all-fibre-coupled” and we are working on integrating this imaging system with a robot arm to fully exploit its “in situ” capability (similar to the point measurement THz probe in the Picobot, Ref 49 (Sci. Rep. 15, 4568 (2025))). Regarding the term “real-time imaging”, we’d like to point out a frame rate of 7 fps has already been accepted as real-time in the THz imaging community as shown in Ref. 28 (Nat. Commun. 11, 2535 (2020)). We demonstrated that our system outperforms the previous system (with the same masks and undersampling ratio) as shown in Fig. 4h in the main text. Compared with the results reported in Ref.28, we improved the frame rate to 14.2 fps.

Finally, the terms “in vivo” “in situ” and “real-time imaging” hereby are not assumed to be applicable to internal body operation. Due to the attenuation properties of the THz light, it is widely agreed that it more suitable for skin/surface detection.

1.2. Engineering Integration Over Core Innovation. Key components are taken from existing technologies:

(1) DMD + silicon wafer for optical THz modulation is a common design.

(2) Hadamard encoding and [1 -1] masking are conventional single-pixel imaging methods.

(3) ATR structures are not new to THz applications.

(4) Image reconstruction uses basic normalized linear superposition, lacking advanced compressive sensing or deep learning approaches.

The performance improvements (2–7 fps frame rate, ~100 μm resolution) stem from component optimization, not fundamental methodological or physical breakthroughs.

Answer: As explained above, the key point (and breakthrough) shown in the manuscript is that this system is all-fibre-coupled. Selecting and optimising the correct combination of existing technologies to realise the all-fibre-coupled design for the SPI application itself is a significant achievement. We’d like to point that the realisation of the wanted

design is very challenging rather than straight-forward. It requires deep knowledge of core physics as well as dexterity with optics.

For example, even the selection of THz polarization needs intensive theoretical derivation of the formulas as shown in the SI and careful theoretical calculations and experimental verification as shown in Fig. 2 in main text.

Again, regarding the image reconstruction, it does not use basic normalized linear superposition approach. Instead, it is reconstructed by multiplying a reconstruction matrix which is obtained by Fourier regularization of the measurement matrix with the L2-norm constraint of the usual compressed sensing minimization problem as shown in Ref. 52 (Opt. Express 26, 20009 (2018)) and 28 (Nat. Commun. 11, 2535 (2020)). This approach first calculates the inversion matrix and stores it in memory. The multiplication of the stored inversion matrix allows a very fast image reconstruction with image quality comparable to standard minimization problems. And this approach outperforms the usual TV-minimization algorithms, which are too slow for real-time display.

We appreciate the deep learning approaches mentioned by the reviewer. Indeed, this advanced approach is planned in our future work. This work explicitly focuses on the most urgent task to accomplish the proof-of-principle design of the hardware.

1.3. Limited Imaging Validation and Clinical Relevance

Despite the authors demonstrated the method by imaging porcine tissues and human skin wounds:

- (1) Sample diversity is insufficient to represent broad biomedical scenarios.
- (2) Results are qualitative; no quantitative comparison exists against clinical standards (e.g., OCT, infrared thermography, ultrasound).
- (3) No histopathological registration confirms imaged structures' physiological significance.
- (4) Statistical evaluations (SNR, CNR, recognition rates) or cross-sample comparisons are absent.

Answer: We agree that we have not presented a comprehensive study of the biomedical samples – this would be a separate paper. The work presented focuses on the proof-of-principle work of the all-fibre-coupled design. Thus, porcine tissue and human skin were measured to verify that the probe developed works as expected rather to demonstrate clinical relevance. We agree with the reviewer that the sample diversity, quantitative comparison, histopathological registration and statistical evaluations are very important. But these elements themselves are better be a separate paper.

The above justification is supported by Reviewer 3, **“The authors also present rudimentary measurements on biological samples, which round the paper off nicely”**.

1.4. Low Modulation Efficiency Limits Performance
Supplementary material reveals a modulation efficiency of ~12%, with only 21% of initial blue light energy reaching the silicon wafer due to losses in fibers/lenses. This low signal contrast necessitates strong normalization and background suppression, potentially compromising imaging depth and stability under low-SNR conditions.

Answer: Losses in optical fibres and lenses are inevitable. By looking at the physics and optics carefully we have managed to design a system which works well and balances all

the physical processes involved, demonstrating that our approach actually works physically. Arguably, 12% is a nearly optimal trade-off between signal-to-noise ratio and imaging resolution/speed. If we had a modulation nearer to saturation (e.g. 90%), the switch off time for that signal is much longer than the charge lifetime (i.e. just because we are nearer to saturation, the drop to 50% signal takes much longer than the next 40% drop to 10%). This would lead to worse imaging resolution and speed. Hence, if we wanted to have better signal-to-noise ratio, this is possible by increasing lifetime and/or optical intensity, but it is always an inherent trade off with resolution and speed. To further illustrate our system's capabilities, we have added in more experimental results to showcase the resolution and contrast as suggested by Reviewer 3.

In conclusion, this work demonstrated solid engineering capabilities in system implementation and offers a reference for THz imaging integration. However, its contributions remain confined to optimizing existing components rather than delivering field-leading innovation. Coupled with misleading terminology, overclaims, narrow application validation, and overstated clinical significance, the manuscript falls short of Nature Communications' standards.

Answer: We sincerely thank Reviewer 1 for the detailed suggestions and comments on our manuscript. Having clarified the misunderstanding regarding our imaging design and explained the reasoning behind the used terminology, we hope that Reviewer 1 now appreciates the significance of our work.

Reviewer #2 (Remarks to the Author):

Dear Authors,

I was asked to review your manuscript NCOMMS-25-44759 "All-fibre-coupled terahertz single-pixel imaging for biomedical applications".

In the manuscript you describe the results obtained with a fibre-coupled terahertz single-pixel imaging system.

A terahertz source is spatially modulated using an optically controlled silicon SLM.

The telecom laser for the terahertz source and detector is fibre coupled and the THz frequency is imaged using TPX lenses, the blue light, used for modulation is fibre coupled using an imaging fibre.

Your work improves on previous experiments in two ways:

1. The shorter lifetimes of carriers in your silicon SLM allows for faster modulation and ultimately higher frame rates.
2. The demonstration of an all fibre coupled experimental setup paves the way for integration of the method onto handheld devices.

I am convinced that your evaluation of your data is valid and your collected data is robust and I can't find any technical flaws with your manuscript

However, I want to recommend to comment on measurement uncertainties in your manuscript.

As far as I understand, most of your measurements are averaged results, it should be straight forward to include confidence intervals/standard deviations in your plots in figures 3-5.

Your work improves on previous results by an increased frame rate and integrating your setup in an all fibre-coupled manner, which means the inclusion of an imaging fibre. The telecom fibre for the THz source and detector were already fibre coupled in previous work.

The core methodology of your experiment, however, stays identical.

I still want to argue that advancing technology in this way is still significant, especially in applied sciences, where the step from the lab into the field often can be challenging.

Your presentation is very clear and easily understandable.

Answer: Thank you for your positive comments. We also appreciate your acknowledgment of the significant achievement of the successful demonstration of this all-fiber configuration.

I only have a few suggestion for improvements on your manuscript.

2.1. As already mentioned a more detailed analysis of errors.

Answer: We thank the reviewer for suggesting the error analysis. We analysed the errors and added them to Fig. 3 d-e and Fig.5 d-f. The errors in Fig.4 g and h are smaller than the symbol size and are therefore not shown. We'd like to point out the oscillations shown in Fig. 4g arise from the different subsets of measurement masks (they do not stem from measurement uncertainty). We use Paley type-I Hadamard matrix to ensure all masks exhibit similar spatial frequencies and thus to enable an easy selection of a subset masks for undersampling. However, this does not guarantee all the masks have the same spatial frequencies, which results in the possibility that higher sampling ratio leads to lower structure similarity index.

In section "Imaging of cartwheel sample" we modified the manuscript to include the error analysis, which reads: "The solid lines in Figs. 3d-3e represent the mean values, and the shaded regions denote the standard deviations across repeated measurements." In section "Undersampling and imaging speed", we added the error analysis, which reads: "Measurements were repeated and standard deviations were found to be smaller than the symbols in Figs. 4g-4h. We note that the oscillations shown in Fig. 4g arise from the different sets of measurement masks rather than measurement uncertainty. Although the Paley type-I masks exhibit similar spatial frequencies, they do not show the same spatial frequency. This leads to lower SSIM for higher sampling ratio, i.e., the oscillations." In section "Imaging of porcine tissue" we added the error analysis, which reads: "The solid lines in Figs.5d-5f represent the mean values, and the shaded regions denote the standard deviations." The indications of the errors are also presented in the captions of Figs. 3-5.

2.2. You are not comparing the structural similarity (SSIM) with your previous work. I would argue that SSIM and FPS/PPS play an important role when benchmarking your experiments against each other.

Answer: Thank you for pointing this out. Compared with our previous work, we achieved a comparable SSIM, indicating the robustness of the undersampling. We got a value of SSIM of 0.85 at sampling ratio of 1 (full set of masks) in the current paper and 0.81 in the previous work – we have added the following information in.

“The SSIM values (0.85, with a full set of masks) are comparable to our previous results (0.81)[28]. Measurements were repeated and standard deviations were found to be smaller than the symbols in Figs. 4g-4h. We note that the oscillations shown in Fig. 4g arise from the different sets of measurement masks rather than measurement uncertainty. Although the Paley type-I masks exhibit similar spatial frequencies, they do not show the same spatial frequency. This leads to lower SSIM for higher sampling ratio, i.e., the oscillations.”

2.3. Fig 5c seems to be taken at lower resolution. Why is that?

Answer: We thank the reviewer for the careful observation. The blurring shown in Fig. 5c arises from a spatial Fourier filter we applied to the image. We revised the figure by removing the filter. The revised figure is shown in revised manuscript.

2.4. Formatting of formulas in the supplement is often unconventional (\times is often used).

Answer: Thank you for spotting this, we have checked this and corrected it.

Otherwise, I have no concerns for your manuscript being published in Nature Communications.

Answer: Thank you very much for your positive feedback and constructive comments.

Reviewer #3 (Remarks to the Author):

The authors present an interesting article on the intriguing combination of multiple techniques for a useful THz imaging system. The authors also present rudimentary measurements on biological samples, which round the paper off nicely. The methodology seems sound and the information provided is sufficient.

3.1 However, the structure of the text does not correspond to that of the journal. The 'Results' section contains a great deal of detail about the setup and methods, which should be found in the 'Methods' section.

Answer: We thank the review for the careful observation. We deleted this section and merged it mainly into the section “Materials and methods”.

3.2 The discussion of the effect of THz polarisation on resolution anisotropy is not fully convincing and is not supported by references, simulations or other evidence. Without this, it is merely an assumption and, as such, should not be discussed in such detail.

Answer: We thank the reviewer for the suggestion. We re-calculated the resolution using a knife-edge sample. We found that the resolution of the horizontally and vertically aligned edge is different. The resolutions are 360 and 660 μm respectively for the horizontally and vertically aligned edges. However, this difference may not only arise from polarization effect. It can also arise from the reflection configuration. Since the THz light

obliquely enters and exits the silicon wafer, the enter and exit positions exhibit a horizontal side shift (see the inset of Fig.1 b), which may worsen the horizontal resolution. In the revised main text, we have justified the details of our system's capabilities with the knife edge results. The new section reads: "We also fabricated a knife-edge sample to estimate the resolution, signal-to-noise ratio and image contrast to characterise the performance of the system. The results (Supplementary Fig. 5 & 6) and the method are presented in the Supplementary Information.

In the revised SI, we have updated the discussion of the polarisation effect on the resolution, which reads: "The different resolutions for the horizontally and vertically aligned edges may arise from the effect of THz polarisation and the reflection configuration. Due to the reflection configuration, the THz light obliquely enters and exits the silicon wafer. The enter and exit positions exhibit a horizontal side shift (see Fig. 1), which is likely to worsen the horizontal resolution."

3.3 The core achievement is the ability to image at high resolution and high measurement rates. While the resolution target presented provides a rough idea of the imaging capabilities, some aspects are unclear.

- Resolution: The measurements with the resolution target (Siemens star) indicate varying resolution throughout the image. Given the provided data, this could be due to the orientation of an edge (similar to astigmatism) or the position within the field of view.
- Contrast: The achievable contrast in this image is not provided. It is stated that the background is zero, which would result in infinite contrast; this is certainly not the case.
- Homogeneity: The images of the resolution target show a highly non-homogeneous image in terms of contrast, signal strength and resolution. While this is not surprising, it needs to be quantified. Is it possible to partially compensate for this inhomogeneity by calibrating each pixel?
- Noise: The image noise has not been quantified. This can occur between neighbouring pixels that are expected to have the same signal strength, as well as between subsequently recorded images.

I suggest applying the razor blade method as a simple experiment to quantify the above values. Place a razor blade, or another sharp, well-defined edge, at the sample plane, then take images at different blade positions. Rotate the blade by 90 degrees and repeat. The images can then be compared with the well-known and simple ground truth and the values above can be extracted. I am willing to help if the authors would like to discuss this in more depth.

Answer: Thank you very much for this suggestion. We have fabricated a knife-edge sample to estimate the resolution, contrast and signal-to-noise ratio (SNR). We deposited a layer of gold to cover half of a silicon wafer. We first obtained THz images with a bare silicon as a reference image. Then we imaged the knife-edge sample. The image of the knife-edge sample is normalised with the reference image to mitigate the adverse inhomogeneity effects arising from the THz beam and the blue light. The resolution is estimated with the profile of a line perpendicular to the edge. The distance related to the 10–90% step height in the profile is assumed to be the resolution. This method is much more precise than barely viewing the image colour variation at the edge. The estimated

resolution for a horizontally aligned edge is 360 μm and 660 μm for a vertically aligned edge.

Regarding the homogeneity, the reviewer also agrees that it is common for an image to exhibit non-homogeneity. To mitigate the non-homogeneity, the images presented in the manuscript are compensated by normalising the images with a reference image obtained with a bare silicon (functions as a metal mirror due to total internal reflection). The detailed normalisation method is presented in the SI section “Image normalisation”. And Supplementary Fig. 4 compares the results with and without normalisation, clearly showing the obvious homogeneity improvement after the normalisation.

The SNR is also estimated with an image of the knife-edge sample. We selected the brightest and darkest regions in the image. The ratio of average value of the brightest region to the standard deviation of the darkest region is assumed to be the SNR. The estimated SNR is 41.

The brightest and darkest regions are also applied to estimate the contrast. We assume that the contrast is the ratio of the difference average values between these two regions to the sum of the average values. The estimated contrast value is 0.7.

We added a new section “Resolution, signal-to-noise ratio and image contrast” in the supplementary material for the results discussed above. The following two figures (Supplementary Fig. 5 & 6) summarise the results of knife-edge sample, which are copied from the mentioned added SI section.

Supplementary Fig. 5 Image resolution estimation. Images of a horizontally aligned knife-edge sample at different positions (a & b). Images of a vertically aligned knife-edge sample at different positions (c & d). The second row (e-h) shows the profiles of the white lines in the first row. The red circles are the experimental data, and the solid lines denote the fitting result in e-f. Fitting parameters x_0 and σ are also presented in each panel.

Supplementary Fig. 6 Signal-to-noise ratio and image contrast. The brightest and darkest regions are applied to estimate the Signal-to-noise ratio and image contrast.

3.4 Other comments:

Line 103. What does 'short' refer to? Please provide an order of magnitude.

Answer: The short lifetime refers to the off-the-shelf silicon wafers' carrier life, which is of the scale of several microseconds. We added a notation to indicate this scale, which now reads: "Off-the-shelf silicon wafers without passivation typically exhibit short carrier lifetimes (several microseconds)".

Line 128 and following: This focuses on the setup and details rather than the results.

Answer: We revised this section and merged it mainly into the section, "Materials and methods".

Line 163 and following: This paragraph seems to explain fundamental optics; I suggest shortening it drastically.

Answer: We thank the reviewer for the suggestion. We revised this section and made it more concise. In the previous version, we discussed both the polarisation and power transmission. In the revision, we deleted the discussion of the polarisation as we agree with the reviewer that this part is fundamental and should be clear for most of the readers. We also shortened the remaining discussion of the power transmission. We'd like to point out that the discussion of the power transmission is necessary as it is important for the polarisation selection and prism design. In addition, the calculation of power transmission is quite complicated as shown by the lengthy derivation presented in the SI.

Line 179: Why 'also'?

Answer: The THz light can be decomposed into ordinary and extraordinary waves in terms of the crystal birefringence. It can be also decomposed into *s*- and *p*-polarized components in terms of total internal reflection. As both the birefringence and total

internal reflection can lead to ellipticity change, we discussed them together by using “also”.

In the revision, this part is deleted as we agree with the reviewer that this part is fundamental and should be clear for most of the readers (see the above reply).

Line 206 and following: Why are both polarizations discussed in detail when the experiment uses just one of them?

Answer: As shown in Fig. 2a, a *s*-polarised THz experiences a transmission loss at the quartz-silicon interface while the *p*-polarised component undergoes no loss. As a result, a *p*-polarised component is more suitable for imaging and therefore *p*-polarisation is selected. We revised the manuscript to clarify this point, which reads: “Notably, at the quartz-silicon Brewster angle, *p*-polarised light undergoes no transmission loss, favouring the imaging application. As a result, *p*-polarised THz is selected in this work, and the prism’s base angle is designed to match the quartz-silicon Brewster angle.”

Line 297: The e-field is certainly not zero.

Answer: Thank you for pointing out this mistake! We have revised the sentence to be: “All images show high absolute THz signal values in the metal cartwheel regions, while the quartz regions exhibit very low absolute THz signal values due to the Brewster angle.”

Line 299: See above.

Answer: We updated the discussion with the resolution experimental results and added the discussion of the possible resolution effect by the horizontal side shift (see the reply above)

Fig. 3c). The plots have the unit 'electric field', but this is imprecise because it is not clear where this electric field is located. I believe the authors plotted the receiver signal, which is not fully linear in the electric field due to the receivers' frequency-dependent roll-off. Furthermore, the setup will have some frequency dependence due to the optics involved and their adjustment. Therefore, for d) and e), I suggest using the term 'receiver signal', and for a to c, I recommend normalising to a full metal mirror as the sample and using the term 'image contrast' or similar. The same applies to other pictures later in the manuscript.

Answer: Regarding the axis label “electric field”, we agree with the reviewer that receiver signal is not fully linear in the electric field due to the receiver’s frequency-dependent roll-off and the setup has some frequency dependence due to the optics involved and their adjustment. We use “receiver signal” to describe the signal as received (Fig. 2b, Supplementary Fig. 3a, Supplementary Fig. 4a and 4b) and “normalised THz signal” for the images and signals normalised with Supplementary Eq. 16. We modified the main text to indicate how the axis are labelled, which reads: “We call the resulting intensity the normalised THz signal (see Supplementary Eq. 16) and the unnormalised one the receiver signal, for example, the signal shown in Fig. 2b”.

Regarding the normalisation, all the images shown in the manuscript are normalised with a reference image obtained with a bare silicon wafer. The normalisation method is

described in SI section “Image normalisation”. And Supplementary Fig. 4 compares the results with and without normalisation, clearly showing the obvious homogeneity improvement after the normalisation. We’d like to point out that a bare silicon functions as the metal mirror mentioned by the reviewer. As the system is based on total internal reflection, a bare silicon wafer well reflects the THz light. Thus, a real metal mirror is not needed to obtain the reference image.

Line 490 and following: Please revise the naming of the components used. If you are unsure, please consult the manufacturer for the correct terminology.

Answer: Thanks for the comment. We have revised the component names.

Response to reviewers

Reviewer #1 (Remarks to the Author):

The authors addressed most of my concerns. However, it is recommended to provide some quantitative data to support the claim that '12% is a nearly optimal trade-off between signal-to-noise ratio and imaging resolution/speed'.

Answer: Thank you for the suggestion to include some quantitative data to support our comment in the previous response to reviewer document about the modulation depth. The modulation depth is inherently linked to the lifetime of the carriers: a longer lifetime will increase modulation depth and improve signal to noise, but at the cost of imaging time, while a shorter lifetime improves imaging speed but at the cost of signal to noise. The fastest switching rate of the DMD is 20 kHz with period time of 50 μ s. With the same laser fluence applied in this work, we achieved modulation depth up to 60% using silicon wafer with lifetime of 300 μ s at the cost of lowering the DMD switching rate, thus, the imaging frame rate. Besides, longer lifetime also increases carrier diffusion length, worsening the resolution (Sci. Adv. 2016; 2 : e1600190). The 12% modulation depth (corresponding to a 7 μ s carrier lifetime) allows us to utilise the fastest DMD switching rate and high imaging resolution. On the other hand, a shorter lifetime would not result in a faster imaging speed (limited by the DMD switching rate), but would only be detrimental to signal to noise - this is why it is near optimal. A full investigation of this complex behaviour would require a modulator with a controllable lifetime - something we are currently working on but not able to do at this moment.

To convey these points in the paper we have added the following into the paper:

"While this may seem like a low modulation depth, it works well in practice because the switch off time is shorter than if a higher modulation depth were used. To increase the modulation depth we would need to use a wafer with a longer carrier lifetime, this would improve signal to noise, but at the cost of imaging time and poorer resolution. Therefore using the short lifetime of 7 μ s allows us to utilize the highest DMD switching rate to achieve high imaging speed. It also gives rise to high imaging resolution, representing a good trade-off between modulation depth, imaging speed, and resolution."

Reviewer #2 (Remarks to the Author):

Dear Authors,

thank you for addressing my concerns with your first manuscript.

I feel that any issues I raised have been resolved and have no concern for your manuscript to be published in Nature Communications.

Answer: We thank the reviewer for recommending the publication of this paper.

Reviewer #3 (Remarks to the Author):

Thank you to the authors for the revised manuscript and for addressing all of my

comments so comprehensively.

Thank you also for the additional knife-edge measurements. I believe these are highly valuable for evaluating such an imaging system. The methodology is sound, and based on the provided data, the obtained values for resolution, SNR and contrast seem plausible. While the interpretation of the anisotropy of the imaging resolution (astigmatism) does not provide a definitive answer, it does offer possible explanations. It would have been great to see a calculation or simulation of this effect, since resolution is one of the key factors of any imaging system. However, this could be the subject of a follow-up paper or letter in a more technical journal.

I have no objections to the publication of this manuscript.

Answer: We thank the the reviewer for appreciating the last response. Regarding the anisotropy of the imaging resolution, we currently suppose it is the from the convolved effects of THz polarisation and the reflection configuration. We agree with the reviewer that resolution is a key factor of an imaging system. However, it takes time to disentangle the convolved effects. We are working on this, the results will be presented in a follow-up paper.